# Cognitive Test Scores and Progressive Cognitive Decline in the Aberdeen 1921 and 1936 Birth Cohorts

**DOI:** 10.3390/brainsci12030318

**Published:** 2022-02-26

**Authors:** Lawrence J. Whalley, Roger T. Staff, Helen Lemmon, Helen C. Fox, Chris McNeil, Alison D. Murray

**Affiliations:** 1Institute of Applied Medical Sciences, Foresterhill, University of Aberdeen, Aberdeen AB25 2ZD, UK; drhelenfox@hotmail.com; 2Aberdeen Royal Infirmary, Foresterhill, Aberdeen AB25 2ZN, UK; r.staff@abdn.ac.uk; 3Royal Cornhill Hospital, Cornhill Rd., Aberdeen AB25 2ZH, UK; helen.lemmon@abdn.ac.uk; 4Aberdeen Biomedical Imaging Centre, Lilian Sutton Building, Aberdeen Royal Infirmary, Aberdeen AB25 2ZN, UK; c.mcneil@abdn.ac.uk (C.M.); a.d.murray@abdn.ac.uk (A.D.M.)

**Keywords:** Mild Cognitive Impairment (MCI), cognitive test, dementia, comorbidity, depression, cognitive reserve, childhood IQ, longitudinal study, normative data

## Abstract

The Aberdeen birth cohorts of 1921 and 1936 (ABC21 and ABC36) were subjected to IQ tests in 1932 or 1947 when they were aged about 11y. They were recruited between 1997–2001 among cognitively healthy community residents and comprehensively phenotyped in a long-term study of brain aging and health up to 2017. Here, we report associations between baseline cognitive test scores and long-term cognitive outcomes. On recruitment, significant sex differences within and between the ABC21 and ABC36 cohorts supported advantages in verbal ability and learning among the ABC36 women that were not significant in ABC21. Comorbid physical disorders were self-reported in both ABC21 and ABC36 but did not contribute to differences in terms of performance in cognitive tests. When used alone without other criteria, cognitive tests scores which fell below the −1.5 SD criterion for tests of progressive matrices, namely verbal learning, digit symbol and block design, did not support the concept that Mild Cognitive Impairment (MCI) is a stable class of acquired loss of function with significant links to the later emergence of a clinical dementia syndrome. This is consistent with many previous reports. Furthermore, because childhood IQ-type data were available, we showed that a lower cognitive performance at about 64 or 78 y than that predicted by IQ at 11 ± 0.5 y did not improve the prediction of progress to MCI or greater cognitive loss. We used binary logistic regression to explore how MCI might contribute to the prediction of later progress to a clinical dementia syndrome. In a fully adjusted model using ABC21 data, we found that non-amnestic MCI, along with factors such as female sex and depressive symptoms, contributed to the prediction of later dementia. A comparable model using ABC36 data did not do so. We propose that (1) MCI criteria restricted to cognitive test scores do not improve the temporal stability of MCI classifications; (2) pathways towards dementia may differ according to age at dementia onset and (3) the concept of MCI may require measures (not captured here) that underly self-reported subjective age-related cognitive decline.

## 1. Introduction

### 1.1. Background

Age-related progressive cognitive decline is a frequent companion of very old age. The symptoms and signs of early Alzheimer’s disease (AD) arise in the context of lifelong trajectories of normative physical and psychological changes, but it is the progressive nature of early AD that distinguishes it from the relatively slow progression of benign, age-related cognitive decline. In broad terms, the early features of AD are of two principal types: they may either impair declarative memory (the recall of specific facts or experiences) and are termed ‘amnestic’ or they concern other cognitive domains including visuo-spatial ability, mental speed, non-verbal reasoning and attentional processes and are termed ‘non-amnestic’ or, less often, ‘atypical’. The detection of early AD in clinical practice should be straightforward: patients who do not have a clinical dementia syndrome but who are impaired in these cognitive domains without sufficient explanation (for example, acquired physical/psychiatric comorbidity or neurodevelopmental abnormality) are considered to be at high risk of progressive impairment. The fact that early AD is difficult to recognize is illustrated by the uncertainties surrounding the term ‘Mild Cognitive Impairment’. The diagnostic category of ‘Mild Cognitive Impairment’ (MCI) was introduced to improve the appraisal of risk in terms of a patient’s progress towards a clinical dementia syndrome. The concept of MCI evolved gradually [1,2,3,4,5,6] and its subclassification has become accepted in terms of ‘amnestic’ and ‘non-amnestic’ types. The hope that MCI will gain credibility and lose its provisional status remains unfulfilled in the face of sustained attempts to reach a consensus [7,8]. Proposals to improve the validity and reliability of MCI classification have scrutinized proposed MCI diagnostic criteria and questioned the inclusion of requirements such as subjective cognitive loss and the absence of any satisfactory alternative explanation [9,10]. These concerns emphasize the provisional nature of MCI and a need for prospective validation studies [11]. 

There are two sets of non-psychometric criteria for MCI. One is a self-report or the observation by another person (who knew the patient before they were presented for clinical assessment) that cognitive problems have emerged in mature adulthood and since endured to impair the person in some significant way but not to an extent that jeopardizes their capacity for independence. A second set of criteria includes the opinion of the examining physician that there is no alternative explanation sufficient to account for acquired mental impairment. These two requirements are open to subjective misinterpretation for multiple possible reasons. There is also the possibility of systematic bias towards the self-reporting of cognitive impairment in a clinical setting whereas this may be lower in community-based studies [12,13,14,15,16]. 

The predictive validity of MCI criteria has been tested to show that MCI is linked to the enduring reduction of psychometric test scores or to associate MCI with a change in clinical status from cognitively impaired without dementia to cognitively impaired with dementia. The choice of psychometric test is difficult. For example, reports that rely on changes in MMSE scores [17,18] may be inappropriate because the MMSE was designed as a screening instrument and not as a measure of cognitive change, despite it often being used as such. The proposal that there is a wide range of patterns of cognitive decline observed among pathways towards clinical dementia syndromes is now recognized but has not yet been widely tested [19,20,21,22]. 

Rates of cognitive change in late adulthood are subject to major influences from diverse sources, some of which are inter-correlated [23,24]. Comorbid disorders are widely recognized as influential confounders of the classification of age-related cognitive decline [25] and are linked to cardiovascular disease [26], sex [27,28,29,30,31,32,33], years of full-time education [34,35,36,37,38] (though not consistently so [39]), a mentally effortful occupation or lifestyle [40] and physical or leisure activities [41,42].

The complex inter-relationships among the covariables of late life cognitive function present difficult methodological problems which are sometimes confounded by the pervasive effects of low socioeconomic status, poverty and chronic ill-health. In part, these arise because hospital-derived samples of those with age-related cognitive impairments are most often studied, whereas the validity of the MCI concept requires as rigorous an examination, as is required for early-stage AD in representative samples of the general population of older adults at risk of age-related cognitive loss [43].

An alternative approach to the classification of MCI is to set aside all proposed criteria other than a small number of widely used cognitive tests. Here, we address **AIM 1** to establish normative values on a verbal memory test used to identify memory impairment in patients in whom a diagnosis of early AD is suspected. We chose to focus on a verbal memory test because the effects of sex on verbal abilities and memory are well established. Although our focus is on scores derived from Rey’s Auditory Verbal Learning Test [44], we also provide scores from fluid and crystalized intelligence tests. These give a more complete ‘normative’ context in which to place any impairment of verbal memory alongside other cognitive and sociodemographic variables (listed below). **AIM 2** is to explore associations between childhood intelligence, comorbidities, sex and cognitive test scores in late adulthood. **AIM 3** is to use repeated cognitive test scores to examine the stability of the psychometric criteria for MCI knowing that some test scores might be subject to well-known practice effects [45,46]. **AIM 4** is to test the prediction of progress, via any cause, to clinical dementia syndrome in two narrow age range cohorts of a fully adjusted model that includes two MCI subtypes (amnestic and non-amnestic) and other predictor variables already linked to dementia and available in our datasets.

### 1.2. Rey’s Auditory Verbal Learning Test (AVLT)

The AVLT is a multi-trial learning test in which 15 monosyllabic words (List A) are presented in five sequential trials with the free recall of these 15 words immediately after presentation. After these five repetitions of free recall, a second ‘interference’ (List B) is presented in the same manner and the is participant asked to recall as many of these as possible. After the ‘interference’ trial, the participant is asked to recall as many words as possible from List A that have already been presented five times. AVLT scores (total trials 1–5) are known to be affected by age [47], sex [48], educational attainments and IQ [49]. 

### 1.3. Physical and Mental Co-Morbidities

We also examine the frequencies of physical and mental co-morbidities. Many processes might impair cognitive function and here we were guided by an earlier influential study [25]. Confounding is frequently caused by any of the many medical or psychological co-morbidities or concurrent prescription or recreational drug use. It is often demanding (or even impossible) for a supervising clinician to determine the exact contribution of co-morbidities towards a patient’s progression towards a clinical dementia syndrome. Reliance on a clinical history from a patient with known or suspected cognitive impairment presents an obvious difficulty. This concern also arises when a patient’s accounts of undocumented illness episodes are noted (for example, transient ischemic attacks, minor “strokes” or head injury) but in the absence of corroboration; these are difficult to include among the plausible explanations for cognitive impairment. An important consequence of the presence of comorbidities, known or suspected, is that the uneven application of exclusion criteria yields samples of possible MCI-affected participants that are difficult to replicate and who may not be generalizable to the older population of adults at risk of dementia. 

## 2. Method

### 2.1. Study Setting

We derived longitudinal data from two well-characterized birth cohorts recruited at ages of about 64 y or 78 y from 1997 through to 2001. Childhood IQ test data were available from the Scottish Mental Surveys of 1932 or 1947 when all children at school in Scotland born in 1921 or 1936 sat a group-administered IQ test when they were aged 11 ± 0.5 y. Our program constitutes an observational, follow-up study of two narrow age range, non-clinical, volunteer cohorts born in 1921 or 1936. All participants were survivors of the Scottish Mental Surveys of 1932 or 1947 who were living independently in the community in 1997. In summary, Aberdeen children born in 1921 or 1936 participated at the age of 11 (range ±0.5 y) in the Scottish Mental Surveys (SMS) of 1932 or 1947 [50]. All children aged 11 ± 0.5 y at school in Scotland on either 1 June 1932 or 4 June 1947 sat the same group-administered national mental ability test. Test scores were correlated (Pearson’s r > 0.8) among a subsample (*n* = 1000) with the Stanford Binet intelligence test. SMS mental ability scores were retained by birth name, date of birth and a school attended age of 11 ± 0.5 y by the Scottish Council for Research in Education, who, in 1996, generously allowed one of the authors (LW) access to their archives. Participants provided fully informed written consent to their continued involvement in a long-term study with follow-up assessment at intervals of between 1.5 and 2.0 y. This long-term study was approved by the Local (Grampian) Ethics of Research Committee in 1998, and later, during follow up, further approval was granted in 2002, 2006 and 2012.

### 2.2. Sampling 

Procedures are reported in detail elsewhere [51]. With the approval of the Local Ethics of Research Committee, the cooperation of local, SMS survivors were sought among eligible Aberdeen community residents living independently and without known serious illness, including clinical dementia syndromes [52], recent bereavement or severe sensory impairment. Between 1999–2001, prospective participants were invited by letter from their family doctor to join a University of Aberdeen longitudinal study of brain aging and health. More than 75% of those invited agreed and were assessed on up to five occasions at intervals of up to 24 months between 1997 and 2014. Critically, none of these volunteers required help with self-care, washing or dressing. Follow up included face-to-face interviews, phone calls and scrutiny of local health and public records. Whenever possible, participants were recalled for face-to-face assessments. Participants who met diagnostic criteria for a clinical dementia syndrome at time of the first assessment are not included in this analysis.

### 2.3. Demographic and Clinical Assessments 

These were conducted in the Clinical Research Unit of a regional psychiatric hospital. All assessments comprised three sessions with two intervening breaks for rest and refreshment. Total testing times were in the range 3.5 to 4 h. **Session 1** was conducted by a trained research nurse who obtained written informed consent and completed a semi-structured interview recording general well-being and current complaints. Years of education were recorded as differences between starting and finishing formal primary and secondary schooling, or later if participants had progressed to higher (tertiary) education with professional qualifications or university graduation, and occupational history (as “best ever regular job”). Subjective cognitive impairment was unreliably ascertained because many participants were unable to provide satisfactory answers. Typically, they were unable to distinguish among self-attributed age-associated memory impairment, feelings of anergia or uninterest and age-related limitations of function. In light of our failure to acquire reliable estimates for self-reported cognitive impairment and our awareness that the repetition of questions about subjective awareness of cognitive impairment would often be invalidated by an earlier performance in a cognitive test and the acceptable estimates of observer-rated cognitive loss (usually based on the observations of a spouse), we could not include these narrative accounts among the data reported here. Sociodemographic data were recorded systematically. The research nurse recorded scores for the Mini-Mental State Examination (MMSE [17]) as well as history of past and current illnesses and their treatments. Participants scoring below 24/30 on the MMSE were not included in the analysis. After a 15 min rest break, in **Session 2**, the research nurse performed a structured clinical examination which included anthropometrics, venesection, respiratory, cardiovascular measurements (blood pressure sitting and standing on three occasions and pulse), 6 m walk time, balance and chair sit/stand repetitions. The research nurse aimed to complete the Barthel Functional Ability Scale at initial and subsequent assessments, but data were unsatisfactory largely because married male participants relied so extensively on their wives irrespective of any changes in cognitive status. On completion of Session 2, after a 15min rest, the participant completed a large number of self-reported questionnaires (details in [51]) while the research nurse conferred with a specialist old age psychiatrist (LJW) if a dementia syndrome was suspected. A discussion with the research nurse determined whether to proceed to **Session 3** or to follow the participant through routine clinical services with appropriate investigation of a suspected clinical dementia syndrome. In 2014, an expert panel of three authors (ADM, RTS, LJW) reviewed all available follow-up data to assign outcomes in terms of specific clinical dementia syndromes using WHO 10 criteria. A hospital case note review identified volunteers who had received hospital-based diagnoses of earlier episodes of psychiatric disorder. These individuals were retained in the study. More than 90% of ABC36 dementia diagnoses were supported by brain imaging data (molecular and/or structural provided by SPECT, CT or MRI) obtained for routine clinical purposes or as part of the research program (ABC21, ABC36). 70% of ABC21 participants underwent brain imaging exams. A follow up to 2014 identified incident cases of dementia (not otherwise specified), who comprise the “dementia outcome group”, along with those classified as cognitively unimpaired (see cognitive tests below) when last observed at up to their fifth assessment after entry to the study. Incident MCI required a decrease in cognitive test scores adjusted for sex, childhood IQ and education greater than −1.5 SD below the group mean on a specific test at study entry. The −1.5 SD criterion was adopted from a review of relevant literature [6,7,8,9,10,11,12,13,14,15,16]. The category of amnestic MCI required an AVLT score below the −1.5 SD criterion. Non-amnestic MCI required an AVLT score above −1.5 SD and at least two other scores below −1.5 SD for any tests relating to RPM, digit symbol or block design.

### 2.4. Cognitive Tests

In Session 3, after a 15–20 min rest and refreshment, cognitive tests were administered by a trained post-doctoral psychologist in the order of (a) Raven’s Standardized Progressive Matrices (RPM) [53], (b) Auditory Verbal Learning Test (AVLT) [44], (c) Digit Symbol Test (DS), Block Design (BD) [54] and The National Adult Reading Test (NART) [55]. Each test was chosen with specific aims is mind. RPM, DS and BD were used to capture aspects of ‘fluid intelligence’ and NART to estimate ‘crystalized’ intelligence [56] because it provides a good estimate of original intelligence or premorbid IQ [57]. Cognitive test scores were ranked as standardized scores and these rankings were used to classify participants in terms of MCI at study entry. Amnestic MCI required an AVLT score more than −1.5 SD below the mean score at first assessment after adjustment for sex, years of full-time education and childhood IQ (age 11 ± 0.5 y). Non-amnestic multidomain MCI required an AVLT higher than −1.5 SD above the adjusted mean and scores less than −1.5 SD below the mean on any two cognitive tests (RPM, DS or Block) adjusted for sex, years of education and childhood IQ. The stability of the classification of MCI was examined by comparing cognitive test scores at follow-up (assessments 2–5) to the distribution of scores obtained for cognitive tests at first assessment.

### 2.5. Co-Morbidities

Data on co-morbidities were obtained by the research nurse, who recorded any self-reported history of treated illnesses, current medications (prescribed and non-prescribed) and limitations of activities attributed to ill-health. Anticholinergic drugs are a well-recognized confounder of studies of cognitive ability [58]. Medications were coded according to their anticholinergic potency by two experienced physicians [59]. Self-reported co-morbidities were ranked as the sum of any of the following: pernicious anemia, Parkinson’s disease, breathing difficulties, angina, hypertension, diabetes, peripheral vascular disease, transient ischemic attacks, heart attack or stroke (minimum 0, maximum 10). 

### 2.6. Statistical Methods

Cognitive test, SMS mental ability and NART scores were approximated to normal distributions and means and standard deviations are used throughout where appropriate. MMSE scores were not normally distributed, were used only for classification purposes and are included in the sample descriptions for completeness. The analysis was performed in four steps. First, to meet **AIM 1**, i.e., to provide normative data for cognitive test scores for two age groups and to compare sexes in each group, we show how these scores varied by childhood IQ and achieved this using Pearson’s correlation method to estimate the strength of association within sexes between childhood IQ and cognitive test scores in two cohorts born in 1921 or 1936. Childhood mental test scores were adjusted for age on date of testing (either 1 June 1932 or 4 June 1947). We used Linear Analysis of Variance (ANOVA) to compare cognitive test scores in late adulthood between sexes and between cohorts’ test scores using Multivariate Analysis of Covariance (MANCOVA) with and without controlling for childhood mental ability. Second, to meet **AIM 2**, we examined associations between cognitive test score, sex and comorbidities. We tested these associations using parametric and non-parametric methods as appropriate. Third, to meet **AIM 3**, we identified sub-samples in each cohort of incident clinical cases of MCI and dementia and compared these with participants who had remained cognitively normal or who met criteria on study entry for Mild Cognitive Impairment. For comparison with other studies, we provide simple percentages of those whose classification remained stable through follow-up, those identified as having Mild Cognitive Impairment who progressed to a dementia syndrome and those who no longer met MCI criteria. To be consistent with the available literature [2,4,16,19,22], or to meet cognitive test criteria for MCI, cognitive test scores were lower than 1.5 standard deviations (SD) below the mean adjusted for sex distribution, educational level attained and age 11 ± 0.5 y mental ability. We explored but discounted the use of an alternative criterion of 1.0 SD below an adjusted mean because this yielded implausibly large MCI categories. The reliability of the MCI classification was based on (a) identification of those participants whose classification remained unchanged during the observation period; (b) those who progressed to MCI and did not revert to ‘not MCI’ and (c) those who were classified as MCI but reverted to ‘not MCI’. Heterogeneity among MCI–not MCI frequencies within participants tested on three occasions was examined using Cochran’s Q test. Changes in MCI classification between successive assessments were tested using McNemar’s χ^2^ test for related samples. To meet **AIM 4**, we explored possible contributions made by MCI subtypes (amnestic and non-amnestic) towards progression towards a clinical dementia syndrome (comprising all causes of dementia) using multinomial logistic regression with all-cause dementia as the dependent variable and using the following covariates: age at first testing (W1), IQ age at 11 ± 0.5 y, frequency of depressive symptoms (Hospital Anxiety and Depression Scale [60]) and categorical covariates of sex, MCI (meeting criteria for either amnestic or non-amnestic MCI), socioeconomic rank, personal history of heart disease, history of hypertension and self-report or hospital case note derived history of treated depressive illness. SPSS v25 was used throughout the analysis. Significance tests were two-tailed throughout.

## 3. Results

### 3.1. Sampling

We set out to provide age and sex stratified normative data for standardized cognitive tests and to show how these cognitive scores varied by sex, level of education and childhood IQ. Table 1 addresses these normative relationships in the necessary detail. We demonstrate the representative nature of our samples. Life course experiences provide the context for our study; these differed between ABC21 and ABC36, and these major influences are probably reflective of developments in nutrition, health care, education and employment opportunities in many developed countries during the twentieth century. In volunteer based non-clinical samples, influences are additional sources of bias. Among those born in 1921, 2792 individuals were tested in an Aberdeen school in 1932. In 1997, 354 of these individuals were traced using the Community Health Index number in Grampian (the region around Aberdeen), 324 were invited to participate and 275 agreed. Family doctors excluded those with known dementia, current serious illness, bereavement or severe sensory impairment and did not provide excluded participants’ details. Data provided at the first assessment were complete for 232 individuals who comprise the Aberdeen Birth Cohort 1921 sub-sample (ABC21). Among these 232 individuals, 128 provided all information required for this study. Of those born in 1936, 2620 individuals were tested in an Aberdeen School in 1947. In 1999, 664 of these individuals were traced using the Community Health Index number in Grampian, of whom 647 were invited to participate and 501 agreed. Out of those who agreed, 480 individuals comprise the Aberdeen Birth Cohort 1936 (ABC36) and of these 425 provided all information required for this study.

This non-clinical, general population sample of volunteers comprised almost all children born in 1921 or 1936 within 5 Km of the center of Aberdeen who were at school in Aberdeen in June 1932 or June 1947 with, for the most part, family origins in Scotland. All volunteers were local community residents, most having lived in Aberdeen since childhood. The Scottish Mental Surveys of 1932 and 1947 [51] administered a valid IQ test to all boys and girls at school on a specific day. These children form complete birth year cohorts of both sexes from one nation and who were for reason of age at risk of dementia. The Moray House Test (MHT) used was validated against the ‘gold standard’ Stanford-Binet IQ Test and our early work established the MHT as a valid and enduring measure of general mental ability [61]. We identified those eligible for recruitment but who declined the invitation from their family doctor. These non-participants had achieved lower scores on the MHT of childhood general mental ability (mean IQ-type scores ABC21, participants: 99.7 ± 13.9; non-participants: 94.5 ± 15.4, F = 9.3, *p* < 0.002. ABC36 participants: 102.7 ± 14.7; non-participants: 95.5 ± 15.3, F = 27.8, *p* < 0.001). Participants who were censored during follow up for reason such as withdrawal of consent, intercurrent illness or death also achieved lower childhood MHT scores than those remaining in the study and were also of lower socioeconomic status and level of education (details available from corresponding author on request).

### 3.2. Sociodemographic and Clinical Data 

Table 1 summarizes sociodemographic and cognitive test scores for the Aberdeen 1921 and 1936 birth cohorts. There are significant differences between men and women among the ABC 1936 birth cohort but not among the ABC 1921 birth cohort. In ABC36, women performed better than men (*p* < 0.001) on verbal memory (AVLT) and mental speed (DS) but less well on visuospatial ability (Block Design). Comparisons of cognitive scores between cohorts showed that among the older cohort, scores were lower than expected in women compared to men on AVLT and on mental speed (DS), whereas men were lower than expected on visuospatial ability compared to women. Verbal ability scores among the ABC 1921 men appeared higher in ABC 1921 than in ABC 1936 and this observation should be re-tested using follow up data obtained on repeat testing of verbal ability in ABC 1936. When statistical analyses were confined to those participants who had completed all cognitive tests (*n* = 128, ABC21; *n* = 405, ABC36), significant differences between men and women remained among ABC36 and only mental speed (DS) was significantly greater in women than men (men 30.3 ± 10.0; women 34.8 ± 12.4; ANOVA, F = 5.2, *p* = 0.03). Repeat cognitive testing through first, second and third assessments showed an expected improvement between first and second assessments, with a decline towards first assessment levels at the third assessment. We attributed this improvement at second assessment to the effects of practice and/or regression to the mean with some reduction of test anxiety and familiarity with testing procedures. We estimated changes in test performance from differences between the third (W3) and first (W1) assessments (i.e., a positive difference reflects cognitive improvement from first to third assessment. These W3-W1 mean differences in MCI and unimpaired groups are summarized in Table 4.

*Comment*: These data suggest heterogeneity between sexes in age-related rates of cognitive change among the ABC21 and ABC36 birth cohorts. Normative cross-sectional data from older adults with dementia are consistent with these observations [28,29,30,31] but do not establish that these differences arise from either (1) a cohort effect (for example, ABC21 may have had fewer educational and different employment opportunities than ABC36), or (2) from underlying neurobiological processes and related pathologies that might differ between sexes, neither do they establish that cognitive test performance could be linked to these. Longitudinal brain structural and functional imaging studies could distinguish between these possibilities.

The prevalence of MCI in ABC21 was 15/178 (8.4%) at an age of about 77y and was 39/425 (9.2%) at age of about 64y in ABC36. These incidence estimates are broadly consistent with recent meta-analyses [9,62,63,64,65] but not with those data from the Lothian 1936 Birth Cohort study [66,67] recruited elsewhere in Scotland using different sampling methods. We consider the differences between the studies in terms of procedures to adjust cognitive scores for the effects of sex, childhood IQ and education to be the likely source of these differences and unlikely to be linked to secular or geographical differences in terms of health, education or occupation between the Lothian and Grampian Regions of Scotland.

### 3.3. Comorbidities

A guide to major comorbidities is listed in Table 2 but Table 2 should be viewed as indicative and not definitive because Table 2 does not summarize the often complex medical histories of participants; most participants self-reported additional recent treatment episodes of other, often minor, illnesses. These episodes were not mutually exclusive and a clearer guide to the frequency of comorbidities is provided by the ‘comorbidity index’ which summarizes the number of major comorbidities reported by these participants. There were significant differences between sexes among the ABC36 cohort but not among the ABC21 cohort: ABC36 men reported more major comorbidities than women (χ^2^ = 6.5, *p* < 0.05). There were differences between cohorts in the frequencies of comorbidities: ABC36 more often reported multiple comorbidities than ABC21 (χ^2^ = 9.1, *p* < 0.01). Table 2 compares cognitive test scores between ABC21 and ABC36 grouped by frequency of self-reported comorbidities. Linear ANOVA did not find significant differences in the total sample grouped by frequency of major comorbidities. When the analysis was restricted to those participants who completed all four tests at study entry and adjusted for sex and mental ability at an age of 11y, there were no overall significant differences within cohorts (ABC21, Pillai’s Trace = 0.068, not significant; ABC36, Pillai’s Trace-0.016, not significant).

Comment: The frequency of comorbidities among the ABC21 and ABC36 cohorts are consistent with earlier reports of cognitively ‘healthy’ old people at risk of dementia [25]. Differences in comorbidities between these cohorts were anticipated on the grounds of a 15 y age difference [25]; ABC36 participants reported multiple comorbidities more often than the ABC21 participants, and this increase was linked to the ABC36 men reporting multiple comorbidities more often [63]. This may be a direct consequence of (1) family doctors excluding those women already known to be seriously ill from recruitment and/or (2) the effects of ‘hardy survivorship’ on the older ABC21 cohort with an additional effect of those eligible to volunteer being selected by family doctors who knew them to be relatively healthy. An apparent retention of cognitive test performance (a positive W3-W1 difference) is an artifact of those who were relatively more healthy and not cognitively impaired remaining in the study. Table 2 about here.

Comment: Table 2 shows more frequent comorbidities among male participants in ABC36 than among women. ABC21 participants did not show the same sex difference. These observations are probably linked to (1) unavailable differences between sexes when family doctors excluded some potential participants on the grounds of poor health and (2) the differential effects of survivorship such that excess comorbidities in the younger (ABC36) cohort were linked to earlier mortality in men and this was linked to a lower frequency of comorbidities in men in the older (ABC21) cohort.

### 3.4. Classification of Outcomes by Degree of Cognitive Impairment

MCI classification in ABC21 was restricted to 178 participants who completed all four cognitive tests at study entry. In ABC21, 11 met criteria for amnestic MCI and four met criteria for non-amnestic MCI. Of all individuals involved, 163 were classified as “not MCI” and were cognitively normal. Table 3 summarizes cognitive scores at first, second and third assessments for ABC21 and ABC36 grouped as not MCI, amnestic MCI and non-amnestic MCI. Multivariate analysis of variance by MCI subgroup and with IQ at age 11 ± 0.5 y, sex and education as covariates showed significant differences between subgroups (MANCOVA, Pillai’s Trace = 0.372, F = 23.3, *p* < 0.001). Table 4 summarizes MCI classifications of “not MCI”, “amnestic MCI”, “non-amnestic MCI” from first to second and third assessments with final any cause dementia outcomes and cognitive data provided at study entry by participants who progressed to a clinical dementia syndrome (not otherwise specified) before the 2014 dementia census date. The distribution on cognitive test scores and their classification as MCI subtypes shown in Table 4 provide a general overview of the data which were entered into linear binary logistic regression. The Cochran’s Q test showed considerable heterogeneity between assessments with Q values above 23.0. McNemar’s test of the distributions of MCI–not MCI frequencies between successive occasions provided values above 21.0 (*p* < 0.001). 

There is a large number of covariance structures that can be used to model the data summarized in Table 4. Our strategy was to see how well the overall model fits when a specified structure is used. 

Comment: Table 4 shows the relative instability of MCI subgroups based on the distribution of cognitive test scores when used alone without subjective cognitive decline or observer report of decline. Although the number of participants was low in ABC21, the numbers in ABC36 were sufficient to suggest that the MCI classification based on psychometric methods alone is too unstable to merit its use either as a guideline to treatment, a predictor of dementia or as a diagnostic entity in its own right. It is uncertain if the addition of a requirement of cognitive decline either as a self-report (‘subjective’) or from a third party would refine MCI classification sufficiently to justify its continued use. The possibility arises that the concept of MCI is more applicable in clinical samples in whom there are almost always grounds for concern about an individual’s cognitive health and that awareness is shared by those affected, thereby meeting a criterion of “subjective awareness”. Psychometric scores may, therefore, fail to measure those aspects of mental life that support self-awareness of cognitive performance.

These data show that MCI subgroups are not stable either over the age periods about 64 to 68 years or the age period age 76 to 80 years. There are no data from this study to support an association between MCI subgroup and progression to dementia over an extended period of observation from 2002 to 2014.

In order to determine the contribution of MCI subtypes to progression to any cause clinical dementia syndrome, a binary logistic regression analysis was performed on any cause clinical dementia syndrome as the outcome and eleven predictors, as listed in Table 5. Potential contributions by frequency of comorbidities and exposure to anticholinergic drugs were examined but discarded because these did not improve the models. Analysis was performed using SPSS v25. Only participants with all the required data were included in the analysis and this reduced the numbers of those who developed dementia. Missing data appeared to be non-random and were associated with incomplete cognitive test scores linked (as above) with lower childhood IQ, lower occupational status and less education. 

In ABC21, a test of the full model with all 11 predictors against a constant only model was statistically reliable: model fit χ^2^ = 29.8, df = 14, *p* = 0.008; Nagelkerke R^2^ = 0.329; Hosmer–Lemeshow χ^2^ = 4.624, df = 8, *p* = 0.797. The model correctly classified 86.9% of participants. The odds of progress to dementia were increased significantly in those who met criteria for non-amnestic MCI (*p* < 0.01), age 11y IQ (*p* < 0.05), age at baseline assessment (*p* < 0.05) and female sex (*p* < 0.05).

In ABC36, tests of the full model with all 11 predictors proved unreliable: χ^2^ = 11.3, df = 14, *p* = 0.66; Nagelkerke R^2^ = 0.124; Hosmer-Lemeshow χ^2^ = 7.6, df = 8, *p* = 0.47. A re-iterative approach with sequential exclusion of variables did not improve the model.

## 4. Conclusions

In this intensive longitudinal observational study without any intervention, no evidence was provided to support the stability of MCI subgrouping over two time periods in two cohorts born 15y apart and studied in late adulthood. These time points occur at ages of interest for the early recognition of Alzheimer’s disease when the risk of progression to dementia and strategies to prevent or delay dementia onset are most often proposed [11,68]. There are data from other longitudinal studies [2] to suggest that the diagnosis of MCI is of greater utility when applied in a clinical setting and is of less value in general population studies. Recent advances in the conceptualization of psychiatric disorders as disruptions of dynamic systems point towards the development of complex multi-component models quantified over substantial observation periods founded on the type of cross-sectional study frequently reported in the characterization of MCI. These approaches may improve the utility of MCI with greater prognostic value.

At a descriptive level, we showed that baseline cognitive test performance was strongly associated with sex and with original (‘premorbid’) childhood intelligence at the age of 11 ± 0.5 y. These associations are sufficient to recommend that sex and original IQ are required to interpret test scores and that the monitoring of temporal trends of ‘normative values’ (as are available in these data) will be valuable. Confounding by level of education is less clear, and it remains uncertain how variations in historical context should be best accounted for in cohorts whose early life opportunities in education and living conditions differed substantially from later born cohorts. In large part, these topics related to IQ, sex and education are subsumed in proposals that “cognitive reserve” plays a significant role in variations in age at dementia onset [69]. According to the cognitive reserve hypothesis, individuals with greater “cognitive reserve” who facing a similar degree of dementia-related brain pathology compared to those with less “cognitive reserve” possess a capacity to buffer or compensate cognitive functions and so postpone the emergence of cognitive impairment. We have elsewhere attempted to quantify “cognitive reserve” [70], but it may also be a potentially informative approach in later work in the context of understanding trajectories towards and away from clinical dementia syndromes. Education, sex and premorbid mental ability are components of “cognitive reserve” already recognized as confounders of MCI classification. Future developments in the early recognition of AD could include the formal evaluation of individual differences in the effectiveness of “cognitive reserve”, with this linked to delays in onset of cognitive impairments attributable to underlying Alzheimer-type neuropathology. 

Notwithstanding, the data reported here have weaknesses that should be addressed in other longitudinal studies and certainly constrain our interpretation provided here. First, we did not satisfactorily account for temporal variations in cognitive test scores (including practice effects and regression towards the mean) which for reasons unrelated to pathophysiology might change between assessments. We have previously explored potential sources of individual differences in test score changes and recognized that intercurrent events (for example, illness, bereavement or stressful life events) could impair test performance and the team members involved in this study were alert to this source of variation. The inclusion of measures of stress responses to intercurrent events could potentially improve the early detection of progressive cognitive impairment [64]. The research nurse, for example, became aware of how serious life events affected the emotional well-being of participants and how these could have affected cognitive test performance in uncertain ways that were not recorded, and this is a defect in our design. There were numerous examples given to the research nurse by participants of stressful instances that included concern for grandchildren exposed to parents’ substance abuse, household thefts and threats of violence.

Research programs that seek to improve the early recognition of Alzheimer’s disease are informed by prospective studies of at-risk populations. Here, our focus was on a general population sample that was unusually informative because childhood IQ test scores were available. Neurobiological data of other at-risk populations suggest that inter-correlative studies that utilize brain imaging will introduce further necessary but more complex analyses. Their value will rely on lessons learnt from the exploration and testing of the MCI concept. Concerns that possible biomarkers will reflect heterogeneity of cognitive deficits [71] will be addressed as new biomarkers (for example [72,73]) are proposed and evaluated.

Our fourth and final aim was to test the contribution of the concept of MCI to the prediction of progress to dementia. We found that in the older ABC21 cohort, non-amnestic MCI in a fully adjusted model that included sex, childhood IQ, age at baseline and depressive symptoms reliably distinguished between those who did or did not progress to dementia over follow up. We have previously shown in a larger sample drawn from the ABC21 birth cohort that lower childhood IQ can be linked to increased risk of a hospital dementia diagnosis [74], and there is evidence elsewhere of increased dementia risk linked to female sex [75] and the frequency of depressive symptoms and/or history of treated depressive illness [76,77]. Taken together, these observations in ABC21 provide a plausible model of increased risk of dementia. It may be safe to infer that the older (ABC21) cohort who scored below the 1.5SD criterion were affected by age-related dementia pathology to a greater extent than the younger (ABC36) cohort. The choice of criterion could differ between age-cohorts and it may be more appropriate in younger old people at risk of dementia to use a less stringent criterion (for example, <−1.0SD). However, this model did not predict progress to dementia in ABC36. Reasons for this probably include the inadequacy of numbers who received a dementia diagnosis during follow up, but we cannot discount the possibility that among the multiple pathways towards dementia [78] there remains pathways not included in this study that would better predict dementia in younger cohorts of the general population.

## Figures and Tables

**Table 1 brainsci-12-00318-t001:** Sociodemographic characteristics, cognitive test scores and HADS-Anxiety (A) and HADS-Depression self-rating scale scores for the Aberdeen 1921 (ABC21) and 1936 (ABC36) birth cohorts.

	ABC21			ABC36				
Sex Total (*n*)	Men *n* = 130	Women *n* = 102	Correlation IQ Age 11 *n* = 232	Men *n* = 236	Women *n* = 244	Correlation IQ Age 11 *n* = 480	Sex Differences	
Age mean ± SD	77.2 ± 0.8	76.9 ± 0.3	−0.11	64.2 ± 1.6	64.2 ± 1.0	-	ABC21	ABC36
Education			0.32 ***			rho = 0.48 ***	χ^2^ = 3.8, ns	χ^2^ = 0.34, ns
1 minimum	100	67		150	143			
2 intermediate	18	23		46	61			
3 higher	12	12		40	40			
Occupation							χ^2^ = 4.84 ns	χ^2^ = 3.97 ns
1 profession/manager	35	17		65	58			
2 skilled/semiskilled	66	52		110	141			
3 unskilled	29	33		61	45			
Childhood IQ	99.9 ± 15.5	101.6 ± 13.1	--	101.0 ± 15.1	102.5 ± 13.8	--	F = 1.44, ns	F = 3.27, ns
Cognitive scores at study entry (mean ± SD)								
RPM correct	27.5 ± 8.8	27.9 ± 12.6	r = 0.09, ns	36.4 ± 8.3	35.0 ± 8.9	r = 0.570 ***	F = 2.1, ns	F = 6.6 *
AVLT	47.6 ± 14.2	49.4 ± 14.1	r = 0.28 **	41.7 ± 9.4	49.0 ± 8.9 ***	r = 0.15 *	F = 4.9 *	F = 74.4 ***
Digit Symbol	30.9 ± 10.2	32.4 ± 12.4	r = 0.14, ns	41.6 ± 10.7	45.5 ± 11.3 ***	r = 0.504 ***	F = 2.8, ns	F = 15.7 ***
Block Design	19.5 ± 6.9	19.1 ± 8.0	r = 0.116	26.4 ± 8.5 ***	23.0 ± 8.3	r = 0.402 ***	F = 5.9 *	F = 14.7 ***
MMSE	28.4 ± 1.5	28.6 ± 1.4	rho = 0.34 ***	28.8 ± 1.5	28.9 ± 1.4	rho = 0.33 ***	F = 0.04, ns	F = 0.89, ns
HADS-D	3.8 ± 3.3	3.5 ± 2.3	r = −0.09, ns	3.1 ± 2.6	3.0 ± 2.5	r = 0.03, ns	F = 0.04, ns	F = 0.82, ns
HADS-A	4.8 ± 2.7	5.6 ± 3.1 **	r = −0.11, ns	5.9 ± 3.1	5.9 ± 3.3	r = 0.07, ns	F = 5.7 *	F = 15.5 ***

Footnotes Table 1. Significance of between-sex differences * *p* < 0.05, ** *p* < 0.01, *** *p* < 0.001. Abbreviations: Education: **1**-Compulsory minimum: ABC21, 9 y or ABC36, 10 y formal schooling. **2**. Intermediate 1–2 y extra school. **3**. Higher education, 3–9 y extra education or training Education:. (Table 1 continued). **1**-Compulsory minimum: ABC21, 9 y or ABC36, 10 y formal schooling. **2**. Intermediate 1–2 y extra school. **3**. Higher education, 3–9 y extra education or training beyond compulsory minimum. Abbreviations: IQ–Intelligence quotient; RPM–Raven’s Standardized Progressive Matrices; AVLT–Rey’s Auditory Verbal Learning Test; MMSE–Mini-Mental State Examination; HADS–Hospital Anxiety (A) and Depression (D) Scale scores; ns–not significant.

**Table 2 brainsci-12-00318-t002:** Frequencies of some comorbidities in ABC21 and ABC 1936 birth cohorts for participants selected by family doctors as known to be without a clinical dementia syndrome, major sensory impairment, serious current illness or recently bereaved (based on items reported in reference [25]).

	ABC21			ABC36		
Comorbidity	Men *n* = 130	Women *n* = 102	All *n* = 232	Men *n* = 236	Women *n* = 244	All *n* = 480
1. pernicious anemia	0	1	1	3	0	3
2. Parkinson’s disease	0	0	0	4	3	7
3. breathing problems	5	8	13	5	2	7
4. hypertension	45	27	72	47	51	98
5. diabetes	4	7	11	12	8	20
6. pvd	1	2	3	2	0	2
7.stroke or TIA	7	5	12	12	8	20
8. heart problems	39	28	67	32	27	59
comorbidity index no comorbidity one comorbid disorder ≥2 comorbid disorders χ^2^	65	56	121 80 31	114	141	255 141 84 *
	44	36		71	70	

	21	10		51	33	

	0.36	ns		12.0	*p* < 0.01	
Prescribed medications No prescribed drugs Prescribed drugs–not anticholinergic Probable or definite anticholinergic Missing χ^2^	31	26	58 106 68	118	117	236 121 123

	61	44		61	59	

	36	32		57	66	
	2	0		0	2	
	0.8	ns		0.7	ns	

Abbreviations (Table 2): pvd–peripheral vascular disease; TIA–Transient Ischemic Attack. Statistical tests: comorbidity index and use of prescribed drug distributions tested using χ^2^ do not differ significantly between sexes or between cohorts, but the frequency of major comorbidities was greater in ABC36 than ABC21 (* χ^2^ = 9.1, *p* < 0.01).

**Table 3 brainsci-12-00318-t003:** Summary (means ± SD) sociodemographic and cognitive scores at study entry for the Aberdeen 1921 and 1936 birth cohorts classified by MCI status on recruitment at ages of about 64 y or 78 y. Participants were followed up from recruitment (1997–2001) up to 2014.

	ABC21			ABC36		
MCI Classification at Recruitment N	Cognitively Normal 163	Amnestic MCI 11	Non-Amnestic MCI 4	Cognitively Normal 386	Amnestic MCI 27	Non-Amnestic MCI 12
IQ at age 10.5–11.5 y	100.8 ± 13.9	102.3 ± 10.3	102.1 ± 7.2	103.5 ± 13.6	90.8 ± 13.9	108.6 ± 12.3
Sex M:F	91:72	6:5	4:0	192:194	15:12	8:4
education						
basic	125	6	2	299	21	10
intermediate	23	6	2	37	2	1
higher	15	0	0	50	4	1
Cognitive test scores on entry						
RPM	28.3 ± 18.9	22.1 ± 6.8	17.5 ± 4.8	36.3 ± 8.1	37.6 ± 5.9	23.9 ± 9.1 ***
AVLT	49.8 ± 12.8	23.5 ± 3.6	40.7 ± 8.3	36.4 ± 7.0	21.6 ± 5.9	34.7 ± 6.9
Digit Symbol	32.7 ± 10.8	21.8 ± 7.1	12.3 ± 2.2	44.3 ± 11.2	39.4 ± 9.8	32.0 ± 12.9 ***
Block Design	31.5 ± 11.2	19.8 ± 7.1	8.1 ± 8.4	25.0 ± 8.4	27.0 ± 6.8	13.7 ± 6.7 ***
IQ at W1	103.4 ± 12.0	91.2 ± 13.5	75.3 ± 5.9	101.6 ± 13.6	91.9 ± 14.9	66.5 ± 9.1
IQ age 11 ± 0.5 y	100.8 ± 13.9	102.3 ± 10.3	102.1 ± 7.2	103.5 ± 13.6	90.8 ± 13.9	108.6 ± 12.3
All-cause dementias						
None	122	9	1	350	26	11
Any dementia	28	2	3	24	1	1
Lost to follow up	13	0	0	12	0	0

IQ at W1 is a composite test score derived by factor analysis of RPM, AVLT, Digits and Block Designs on study entry and transformed to an IQ-type score (mean 100, SD 15). *** *p* < 0.001.

**Table 4 brainsci-12-00318-t004:** Stability among MCI classifications of “not MCI”, “amnestic MCI” and “non-amnestic MCI” from first to second and third assessments with final any cause dementia outcomes. “Losses to follow-up” were attributed to death, illness or withdrawal from the study.

ABC21						
	First Assessment	Group Change	Second Assessment	Group Change	Third Assessment	Dementia before 2014
	Age (mean ± SD) 76.9 ± 0.4		Age (mean ± SD) 78.2 ± 0.4		Age (mean ± SD) 79.7 ± 0.4	
‘*n*’ totals for assessments first = 178 ‘*n*’ totals for assessments second = 127 ‘*n*’ totals for assessments third = 87						
Not MCI	163	Amnestic (9) lost to follow-up (51)	115	Amnestic (7) Non-amnestic (3) lost to follow-up (30)	75	24
Amnestic MCI	11	Not MCI (9) Amnestic (2) Non-amnestic (2) lost to follow-up (1)	12	Not MCI (3) lost to follow-up (1)	9	2
Non-amnestic	4	to not MCI (3) to amnestic (1)	0		3	3
**ABC36**						
	**First Assessment**	**Group Change**	**Second Assessment**	**Group Change**	**Third Assessment**	**Dementia before 2014**
	Age (mean ± SD) 64.2 ± 1.0		Age (mean ± SD) 66.3 ± 0.9		Age (mean ± SD) 68.3 ± 0.9	
‘*n*’ totals for assessments first = 425 ‘*n*’ totals for assessments second = 352 ‘*n*’ totals for assessments third = 323						
Not MCI	386	Amnestic (18) Non-amnestic (5) lost to follow-up (65)	318	Amnestic (17) Non-amnestic (5) lost to follow-up (35)	277	30
Amnestic MCI	27	Not MCI (18) Non-amnestic (2) lost to follow-up (0)	27	Not MCI (14) to non-amnestic (1) lost to follow-up (13)	28	1
Non-amnestic	12	to not MCI (2) to amnestic (1) lost to follow-up (0)	16	to not MCI (2) to amnestic (3) lost to follow-up (0)	17	1

**Table 5 brainsci-12-00318-t005:** Fully adjusted binary linear regression models of progression to any cause dementia in the Aberdeen 1921 (ABC21) and Aberdeen 1936 (ABC36) birth cohorts using baseline predictor variables. Sample sizes are curtailed by losses to follow up and incomplete baseline cognitive data. There are 28 any cause dementia outcomes in ABC21 and 13 any cause dementia outcomes in ABC36 in the restricted samples reported here.

Predictor	ABC21 (*n* = 123)		ABC36 (*n* = 297)	
	Odds Ratio (95% CI)	Significance	Odds Ratio (95%CI)	Significance
MCI				
not MCI				
amnestic MCI	3.27 (0.75–14.35)	*p* = 0.12	3.45 (0.33–35.9)	*p* = 0.30
non-amnestic MCI	6.05 (1.55–25.58)	*p* = 0.009 **	0.73 (0.43–12.22)	*p* = 0.82
IQ age 11 y	1.02 (0.98–1.06)	*p* = 0.31	0.99 (0.95–1.04)	*p* = 0.78
IQ at W1	0.95 (0.91–1.0)	*p* = 0.03 *	1.10 (0.97–1.23)	*p* = 0.13
Age at W1	0.95 (0.91–1.00)	*p* = 0.03 *	1.1 (0.58–2.33)	*p* = 0.68
HADS-dep score	0.69 (0.51–0.92)	*p* = 0.01 **	1.20 (0.98–1.48)	*p* = 0.08
treated depression	5.0 (0.30–81.9)	*p* = 0.26	2.60 (0.43–15.9)	*p* = 0.30
heart disease	0.93 (0.14–5.55)	*p* = 0.93	1.76 (0.44–7.13)	*p* = 0.43
hypertension	0.72 (0.25–2.10)	*p* = 0.32	1.58 (0.47–5.25)	*p* = 0.46
Occupation				
1. professional/manager	0.81	*p* = 0.80	0.74	*p* = 0.68
2. skilled worker	(0.17–3.89) 0.86	*p* = 0.88	(0.18–3.01) 0.53	*p* = 0.51
3. unskilled manual	(0.14–5.55)		(0.08–3.44)	
Education				
1. basic				
2. intermediate	0.46 (0.10–2.05)	*p* = 0.31	3.17 (0.56–18.1)	*p* = 0.57
3. higher	1.29 (0.21–8.09)	*p* = 0.78	3.6 (0.08–3.02)	*p* = 0.43
Sex male = 1 Female = 2	3.79 (1.20–11.96)	*p* = 0.023 *	1.39 (0.42–4.6)	*p* = 0.59

Footnotes: ** *p* < 0.01; * *p* < 0.05 W1–first assessment; treated depression–self report of prescribed antidepressant or hospital treated episode from case notes. History of heart disease or hypertension from self report. Abbreviations: CI–confidence intervals; HADS–Hospital Anxiety and Depression Scale.

## Data Availability

Access to the ABC21 and ABC36 combined database is managed by the University of Aberdeen longitudinal studies steering committee. Anonymized data is available to bona fide researchers by application to this committee (c.mcneil@abdn.ac.uk).

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
