# Peer review of "Cognitive Test Scores and Progressive Cognitive Decline in the Aberdeen 1921 and 1936 Birth Cohorts"

_brainsci, 2022, doi:10.3390/brainsci12030318_

Round 1

Reviewer 1 Report

Main issue: the aim.

You aimed "(1) to rely only on cognitive test scores to classify MCI;(2) to compare gender effects between two birth cohorts born in...” 

I think the rationale for these aims should be better presented, citing and describing relevant literature. 

First, what is the rationale to focus on verbal abilities? Second, why did you investigate gender differences?

You state before "Here, we first concentrate on establishing normative values on one type of verbal memory test often used to identify memory impairment in patients in whom a diagnosis of early AD is suspected and because gender effects on verbal abilities and memory are well established”, but I think these points merit to be more articulated. 

In addition, you describe a set of cognitive tests. Which is the rationale for using them? On page 4 you briefly state it, but i think you should describe the rationale of using them in the introduction.

In statistical method then I suggest following the same line of reasoning in presenting analyses as exposed in the aims, i.e., I do not understand by now explicitly the association between aims and analyses.

Other points:

- Abstract. “On recruitment, significant gender differences within and between ABC21 and ABC36 cohorts supported advantages in verbal ability and learning among ABC36 women that were not detected in ABC36” This sentence is not clear to me, I suggest to make it clearer.

- Participants. The following aspects are not clear to me:

  • “Subjective cognitive impairment was unreliably ascertained because many participants were unable to provide satisfactory answers. Typically, they could not distinguish among self-attributed age-associated memory impairment, feelings of anergia or uninterest, and age-related function limitations.”
  • Participants scoring below 24/30 on the MMSE were not included in the analysis. 

I suggest that you better motivate and explain these points.

- Discussion. A paragraph discussing the results is missing (only the conclusion paragraph is present). I suggest to insert a discussion that I think should follow the line of reasoning of the rationale of the study (see my main point above). Possibly, I would also add implications of the present results (your last paragraph in the present version is in this direction, but I found it difficult to follow).

Then a conclusion section with a take-home message could be reformulated at the end of the paper.

Minor points (typos):

  • Throughout the paper, I noted some redundant spaces
  • Page 2. [34--38]

Author Response

RESPONSE  TO REVIEWERS’ REPORTS

We thank the reviewers for their careful and detailed evaluation of our work. We are grateful for their suggestions and have re-analysed and rewritten much of our manuscript to reflect their concerns. Reviewer#3 makes a useful suggestion to provide more detail about associations between childhood IQ and late cognitive performance. We did not make these suggested changes partly because we have addressed these in much detail in our previous  publications (also cited here) and also because we felt that the new tables were sufficiently self-explanatory not to merit an illustration to emphasise our main findings.

BACKGROUND LITERATURE

We have considered our choice of cited works and believe on balance that we have identified the most relevant authors (eg Petersen, Ritchie, Stephan) and many of their most highly cited papers. Our purpose was to explore an approach to MCI which relies only on psychometric test scores, and contrasts with earlier work that included more subjective judgements to explain cognitive impairments in late life. Our focus on Rey’s Auditory Verbal Learning Test is based on reasoning that verbal learning is a core feature of amnestic MCI subtype, other tests are included for comparison with non-amnestic MCI subtype also without subjective elements. We feel that our reasoning that leads to our aims is set out in sufficient detail in the introduction and is adequately supported by cited literature and new references now included in the discussion. The suggestion of new citations in the introduction  of a largely psychometric or statistical nature would not, in our view, make the paper more accessible to the reader of a clinical journal though this might be a more complete treatment of these issues when presented in a more specialized journal.

AIMS

We have rewritten the introduction to spell out our aims more clearly and to link these to our revised statistical method. These changes meet Reviewer #1 request that we “explicitly” associate “aims and analysis”.

STATISTICAL ANALYSIS

Reviewer#2 expresses “major concerns” about our lack of a detailed “surrounding  the analysis of the MCI classification”. We have not changed our use of the -1.5SD criterion for MCI because this remains the most frequently criterion in this context but we have accepted the suggestions made by Reviewer#2 to improve our analysis.  We have, in light of this advice, completely re-analysed our dataset and introduced wherever possible formal statistical tests of our hypotheses and deleted the largely narrative approach adopted in the submitted version of our manuscript. The revised tables were produced with Reviewer#2 concerns in mind, the term ‘censored’ is now explained and arithmetical inconsistencies corrected which were attributable to non-random loss from follow up. We apologise for these errors.

The linear binary logistic regression analysis is completely new and, although not novel, pulls all these data together in a way we hope meets Reviewer#2’s concerns about the adequacy of our statistical treatment of these data.

Reviewer 2 Report

In their manuscript, "Early recognition of progressive cognitive decline in the Aberdeen 1921 and 1936 birth cohorts", Whalley and colleagues follow up two (ABC21/36) ageing cohorts of older (age 75+ years) cognitively healthy adults using clinical assessments and cognitive testing at multiple time points. IQ data (tested at age 11 years) was available for all participants because they were participants in the Scottish Mental Surveys of 1932 or 147. This is an invaluable dataset, and the authors are to be commended for so diligently following these individuals. The authors provide an excellent summary of the provenance of these data, and clearly describe how participants were (re-)recruited and followed from ~2000 to 2014.   Whalley et al. describe and summarise the basic demographics, physical (e.g., co-morbidities) and mental health of their samples, and report cognitive scores, as assessed upon entry to the current study, from 4 tasks (RPM, AVLT, DS, BD) and the MMSE. Indeed, these data themselves comprise an important normative dataset.   From these data, the authors report:
  1. Male / female differences in some cognitive within one sample (ABC36), but not the other (ABC21)
  2. Cognitive scores were correlated with IQ at age 11 years.
  3. No differences in frequencies of (self-reported) co-morbidities between genders or samples (and co-morbidities were unrelated to cognitive differences
  4. The numbers of cases classified as "not MCI", "Amnestic MCI", and "Non-Amnestic ACI" at three time points (and the number of cases changing between groups)
  From these results, they conclude that criteria based on (only these) cognitive tests do not improve the temporal stability of MCI classification.   Unfortunately, I have major concerns about the manuscript that prevent me recommending it for publication.   The primary conclusion of the paper -- that "MCI criteria restricted to cognitive test scores do not improve the temporal stability of MCI classification" -- is not supported by any statistical or quantitative analysis other than a subjective interpretation of Table 4 (i.e., "Table 4 shows the relative instability of MCI subgroups based on the distribution of cognitive scores..."). Given the attention to details about data collection, there is a glaring absence of detail surrounding the analysis of MCI classifications. For example, any measurement (especially cognitive measures) includes some error / random fluctuation that can expressed in terms of the measure's reliability, etc. To what extent were the changes in MCI classification driven by noise, or factors such as regression to the mean? A hard threshold of 1.5SDs below the mean on a given test was (arbitrarily?) selected as the criterion for MCI - but without knowing the reliability of these tests (or some other measure of stability), we have no way of knowing: a) if an individual's true score lies below 1.5SDs (i.e., do the error bars around the single measurement overlap the threshold?), and b) whether the change in individuals' scores are actually significant / meaningful. How many of the cases that changed MCI classification shifted from slightly below to slightly above the threshold because of, e.g., regression to the mean? A more rigorous analyses must be performed, using methods such reliable change index, regression-based methods, or others (see references below).   On a related note: basic and critical details about the statistical tests were not provided. How were differences such as gender effects tested? Student's or Welch's t-test? Parametric or non-parametric methods? How were cognitive scores adjusted for age/gender, etc.? How / why was a threshold of 1.5 SDs selected as the threshold for defining MCI?   Additionally, there are considerable errors in the manuscript that made it very difficult to read - I often found myself re-reading sentences or sections or scrutinising tables to try to understand what was being described. For example:
    • Abstract (line 16) - "... supported advantages in in verbal ability and learning among ABC36 women that were not detected in ABC36" . What?? Should this say, "... that were not detected in ABC21"?
    • Lines 178-181 seem important but do not make sense syntactically. "These incident cases ... comprise the dementia outcome group with which those classified as cognitive unimpaired (see cognitive tests below) when last observed at up to their fifth assessment after entry to the study".
    • Line 311 - "Table 3 summarises cognitive scores at the first, second and third assessments..." But as far I can discern, only data for the first assessment are shown in in Table 3.
    • Table 3 - what statistical tests do the asterisks correspond to? That is, what are the cognitive scores for non-amnestic MCI group in ABC36 different from?
    • Table 4 - What does "censored" mean?
    • Table 4 - the numbers don't always add up as I assume that they should. For example, in ABC21 there are 4 cases in the non-amnestic MCI group at the first assessment, and all four change category (1 to amnestic, 3 to not MCI) on the second assessment. Hence there are 0 cases of Non-amnestic MCI at the second asssessment. However, what about the 2 cases that change from Amnestic MCI to Non-Amnestic MCI? Shouldn't there be 2 cases of Non-Amnestic MCI at the 2nd assessment? This table does not make sense, yet is central to the authors' main conclusion.
    Duff, K. et al. Predicting cognitive change in older adults: The relative contribution of practice effects. Arch. Clin. Neuropsychol. 25, 81–88 (2010).   Duff, K. Current topics in science and practice evidence-based indicators of neuropsychological change in the individual patient: Relevant concepts and methods. Arch. Clin. Neuropsychol. 27, 248–261 (2012).   Maassen, G. H., Bossema, E. & Brand, N. Reliable change and practice effects: Outcomes of various indices compared. J. Clin. Exp. Neuropsychol. 31, 339–352 (2009).

Author Response

(The authors gave the same response as above.)

Reviewer 3 Report

The most interesting part is the relationship between IQ at age 11 and cognitive performance in MMSE and AVLT. This might be discussed in more detail.

The data on the changes of diagnostic category at the three assessments could perhaps be presented more clearly than in table 4, perhaps with a diagram?

Author Response

(The authors gave the same response as above.)

Round 2

Reviewer 1 Report

Although you did not reply point by point to my comments, I think the paper has now improved. I have no additional comments.

Author Response

There are no comments that require a response

Reviewer 2 Report

The authors have done a commendable job addressing all the Reviewers' comments. This reviewer feels that the manuscript is much improved, especially in the presentation and clarity of the results. The new analysis -- i.e., predicting clinical dementia using a logistic regression model -- is a nice added touch. For the most part, my concerns have been addressed by clarifications and corrections to the tables of results.

However, I am still concerned that one of the primary conclusions of the paper, that, "... no evidence was provided to support the stability of MCI subgrouping over two time periods...", is too strongly stated; especially as the first conclusion reported in the Discussion. Although Table 4 has been improved and clarified, the authors' conclusion is based solely on an inspection of this table with no supporting statistics or analyses. Even more simply, what are the criteria upon which they have decided that "Table 4 shows the relative instability of MCI subgroups based on the distribution of cognitive test scores..." ? How is this conclusion self evident? Were they expecting to be no changes in classification? Or, only classification in one direction (e.g., no MCI -> MCI)? In order for any claims to be made about the stability of MCI classification from cognitive scores, there must be some sort of analysis or at least a better description about what makes the classification table "appear unstable". If only the latter is possible, then I would suggest that the authors phrase their conclusion more cautiously.

On a related note, I appreciate that the authors include a discussion about limitations  - specifically, temporal variations in cognitive scores. I also propose that the rudimentary method of MCI classification (i.e., a simple 1.5SD threshold) contributes to the "apparent instability" of MCI classification. As discussed in my previous comments, even a miniscule changes in a cognitive score could lead to a change in MCI category if the participant's initial score was ever so slightly above or below the threshold. The approach to MCI classification should be acknowledged as a limitation here, as well.

Similarly, in their new logistic regression analysis why have the authors used MCI classification as a predictor of clinical dementia, rather than the more granular test scores? That is, they could just use z-scores prior to thresholding for MCI classification as independent variables in the logistic regression models. Perhaps the initial cognitive scores would be more informative (e.g., as significant predictors of dementia)?

Minor Corrections:

Section 3.4 - "Table 4 summarizes sociodemographic and cognitive data...", but Table 4 is only the MCI classification data?

Lines 80-82: "First, we did not satisfactorily account for temporal variations in cognitive test scores (including practice effects and regression towards the mean) which for reasons unrelated to pathophysiology might vary over time".  This could be improved to read, "... which for reasons unrelated to pathophysiology will lead to some cognitive scores changing (e.g., appearing to improve) between assessment" - or something like that.

Author Response

We thank the reviewer for the great care taken to assess our work. The reviewer suggests a more cautious approach to interpretation of the main findings and we accept this would strengthen the paper. We have made minor changes to our use of language in the abstract (for example, "propose" not "conclude", "interpretation" not "conclusions" and elsewhere as indicated by the reviewer in the discussion. Changes are also made where suggested in the discussion to temporal variation in cognitive test scores.

There is a more substantial issue concerning our use of the MCI classification in the logistic regression analysis where the reviewer suggests that we consider using the cognitive test scores and not the MCI classification as predictors. We are reluctant to follow this advice because, although interesting, it will not provide a clinically useful test our hypotheses arising from current clinical use of the MCI classification based on adjusted cognitive scores more than 1.5SD below the group mean. The suggestion that we use "granular" cognitive scores is valuable but also does not test our original hypotheses. We regard the analyses presented here that were developed to include linear logistic regression as an interim step before progressing to more advanced statistical models (preliminary path analysis and structural equations) but difficulties working together in current circumstances have not allowed this work to progress as planned. We acknowledge the limitations of our approach in the discussion and while grateful to the reviewer for these suggestions are unable in our present circumstances to make a further major revision with a fresh approach to statistical analysis.